# An Efficient Method for the Intermodal Four-Wave Mixing Process

**DOI:** 10.3390/ma15134550

**Published:** 2022-06-28

**Authors:** Michał Kwaśny, Paweł Mergo, Marek Napierała, Krzysztof Markiewicz, Urszula A. Laudyn

**Affiliations:** 1Faculty of Physics, Warsaw University of Technology, Koszykowa 75, 00-662 Warsaw, Poland; michal.kwasny@pw.edu.pl; 2Laboratory of Optical Fiber Technology, Maria Curie-Sklodowska University, pl. M. Curie-Sklodowskiej 3, 20-031 Lublin, Poland; pawel.mergo@mail.umcs.pl; 3InPhoTech Sp. z o.o., Poznańska 400, 05-850 Ołtarzew, Poland; mnapierala@inphotech.pl (M.N.); kmarkiewicz@inphotech.pl (K.M.)

**Keywords:** four-wave mixing, nonlinear frequency conversion, a few-mode optical fiber

## Abstract

We demonstrate a partially degenerated intermodal four-wave mixing (FWM) process realized in a few-mode nonlinear optical fiber, leading to the effective generation of visible red and blue light from 532 nm sub-nanosecond pulses. We present a self-seeded FWM configuration with a signal beam obtained in the additional section of the same type of fiber that ensures perfect matching between the seed and the Stokes beams. Over 40% of the wavelength conversion efficiency in the FWM process was obtained using a fiber length shorter than 1 m.

## 1. Introduction

Light parametric conversion through a third-order nonlinear four-wave mixing (FWM) in optical fibers is a commonly utilized technique for frequency conversion. This process is described by χ^(3)^ nonlinear susceptibility and, for the first time, was observed experimentally by R. H. Stolen [1]. The FWM process leads to exchanging energy between signal, pump(s), and idler beams only if all the beams involved in a process are phase-matched along with the propagation along with the fiber core. To date, FWM has been extensively studied in various types of optical fibers, including single- [2,3] and multimode ones [4,5,6,7,8], as well as photonic crystal fibers (PCFs) [9,10,11]. A typical method for satisfying a phase-matching condition in PCFs [12] is the utilization of dispersion-shifted fibers designed at the telecom band [13] and the visible range [14]. Such conditions can be relatively easily obtained in PCFs, where the propagation constants β of the guided modes can be designed by changing the holes’ pitch and diameter, resulting in a wide range of β-curves for the guided modes [9]. The phase-matching condition in step-index fibers can be fulfilled by close to the zero-dispersion wavelength pumping [15,16], birefringence matching [17,18], or by utilization of the intermodal four-wave mixing technique (IM-FWM). IM-FWM can occur in a few-mode optical fibers (FMF) and is based on the propagation of the pump, Stokes and anti-Stokes beams in the form of fundamental as well as higher-order modes (HOM), resulting in fulfilling the phase-matching condition. This method makes it possible to obtain significant frequency shifts between optical fields involved in the nonlinear process [8,19]. One of the advantages of the intermodal approach is that the wavelengths of the Stokes and the anti-Stokes beams go well beyond the stimulated Raman scattering (SRS) bandwidth, which is one of the sources of degradation of efficiency of the FWM process [20]. In our experimental observations, the Stokes and anti-Stokes beams within the IM-FWM process can be generated spontaneously from vacuum noise as well as during the stimulated process, where together with an intense pump beam, a weak signal beam of a wavelength matched to the Stokes beam was co-coupled to the core of a short section of nonlinear fiber. In the latter case, a weak signal beam (mostly continuous wave—CW) is amplified along with the propagation distance through a fiber in the presence of an intense pump beam. Such an approach results in a better wavelength conversion efficiency.

In our work, we analyze the Stokes and anti-Stokes beams’ power and FWM process efficiency versus the power of a pump beam in both cases, with and without using an additional signal beam.

## 2. Materials and Methods

The IM-FWM was realized in a graded-index optical fiber with a core diameter of 27 μm and a maximum refractive index difference of 0.008. Such a fiber supports 42 mode groups at a wavelength of 532 nm, which may be treated as a multimode fiber. The effective refractive index and group refractive index values were calculated and are plotted in Figure 1a,b. The mode field diameter of the fundamental mode at 532 nm is equal to 7.6 μm. Zero dispersion wavelength (ZDW) occurs for the LP_01_ mode at 1292 nm, for LP_11_ at 1290 nm, and the LP_02_ and LP_21_ modes at 1306 nm. As presented in Figure 1c, higher-order modes within visible wavelength bands vanish before they reach ZDW. Among others, the phase-matching condition for a pump beam in the fundamental mode at 532 nm is satisfied when Stokes/anti-Stokes beams excite LP_02_/LP_01_ modes at 616.8 nm/467.7 nm or LP_12_/LP_01_ modes at 642.2 nm/454.1 nm, respectively.

The detailed experimental setup is shown in Figure 2. It utilizes a q-switched Nd:YAG laser with a pulse duration of 500 ps, a pulse repetition rate of 1 kHz and a maximum peak power of 110 kW that operates at wavelength λ = 1064 nm. The laser beam was frequency-doubled in a second harmonic generation process using a nonlinear KTP crystal. As a result, an intense beam within a visible band (λ=532 nm, 500 ps pulse width, 1 kHz repetition rate, and 45 kW of peak power) was generated and used as a pump for the IM-FWM process. A half-wave plate (λ/2) and a polarizer were used to control the optical power launched into the fiber. For coupling/decoupling of light, we used a 10X Plan Achromat Objective with 0.25 NA.

Even though the power of the Stokes and anti-Stokes beams should increase along with the fiber length, the recorded beams’ powers are limited, predominantly by the Raman effect and due to temporal walk-off (insignificantly at such short propagation distance and sub-nanosecond pulse duration) [11]. As the fiber length increases, the FWM can appear firstly; however, during further propagation in the nonlinear medium, the Raman effect would lead to the spectral broadening of both Stokes and anti-Stokes beams. On the other hand, the spontaneous Raman scattering in shorter fiber sections would be much less significant, notwithstanding the FWM conversion efficiency would also decrease. We used a 0.8 m section of a fiber under test (FUT) in a straightforward experimental configuration that utilizes a single pass of an intense pump beam. The specific length of the fiber was selected experimentally as a kind of trade-off between efficient IM-FWM frequency conversion to Stokes and anti-Stokes beams and minimal degeneration of all FWM waves caused by the Raman effect.

The FUT was kept straight to avoid external strain and stress-induced birefrin-gence. Thus, we can assume that the polarization of light is preserved during propaga-tion. The fiber was pumped at 532 nm using a free-space beam coupling and mi-cro-translation stage supported by a piezoelectric drive to precisely adjust the position of the fiber. For the purpose of mainly fundamental mode excitation for the pump beam, the position of the FUT was carefully adjusted to get a clean LP_01_-like modal field profile in the linear propagation regime, both in the near- and far-field. Then, for the high-power pump beam propagation, the fiber position was adjusted again to maxim-ize the power of the Stokes/anti-Stokes beams at the output, however, continuously monitoring the modal field profile to keep the propagation of the pump beam pre-dominantly in the fundamental mode.

Among other possibilities, intermodal phase-matching in FUT can be achieved when the 532 nm pump propagates in the fundamental LP_01_ mode. To fulfil this condition, we verified the quality of a Gaussian profile of a beam with a beam profiler (BP209-VIS) and CCD camera that visualizes the light intensity distribution. To ensure high coupling efficiency, we used a 3-Axis NanoMax Flexure Stage to generate a clean LP_01_-like green-color modal field both in the near and in the far-field. Using a microscope objective of NA = 0.25, a beam was focused to a spot of a waist diameter 2w_0_ = 2.8 μm and precisely coupled to a fiber of 0.8 m length. The fiber was kept straight to avoid additional strain and prevent higher-order modes’ excitation, keeping predominantly the fundamental LP_01_ mode. In order to prevent damage to the end of the optical fiber, the average power of a pump beam was limited to 10 mW (a peak pulse power ~20 kW). Depending on the measurement, the output beam was routed between the power meter/optical spectrum analyzer (OSA) and a prism (or set of two prisms) combined with a CCD camera. Such a switchable configuration was utilized for data acquisition concerning the output power, spectral characteristics and modal profiles of the pump, Stokes and anti-Stokes beams.

In the second part of our work, we investigated the seeded FWM process using an additional low power signal beam of a wavelength matched to the Stokes beam that was co-coupled and co-propagated with an intense pump beam. Such an approach minimizes the impact of stimulated Raman scattering on the optical output field. With an additional seed beam, the pump beam is more efficiently converted to the Stokes and anti-Stokes frequencies, as the signal beam is gained from the beginning of the fi-ber, compared to a case with spontaneous generation Stokes/anti-Stokes generation from vacuum noise. As a result, the FWM conversion efficiency significantly increases.

Despite the advantages, the approach with the additional seed beam also has its drawbacks. One is the necessity to use a laser source with a wavelength adjusted to the Stokes or the anti-Stokes waves, which is not a significant problem in the infrared (IR) pumping regime. In a visible frequency range, of which the most common and easily obtainable wavelength is 532 nm (the second harmonic of Nd:YAG laser), the wavelengths of the Stokes and anti-Stokes beams are both in the visible red and blue light range. Thus, matching with the wavelength of a signal beam requires an appropriate, preferably tunable, light source of coherent light. Moreover, a proper modal field distribution that corresponds to the appropriate order mode in the optical fiber should also be ensured. The proper evaluation of the HOM modes number for the generated Stokes/anti-Stokes beams can be realized, using, for example, the S^2^ method [21,22]. The proper determination of the modal field composition of each stimulated mode would then be essential for a selective mode excitation of a signal beam [23,24]. There are works by other authors [25] concerning IM-FWM experiments using a weak CW signal beam that presents the increase in efficiency of the nonlinear wavelength conversion. Some works also present another approach to the self-seeded FWM process [26,27], where the efficient generation of self-seeded FWM utilizes a double-pass technique, which on the other hand, leads to an increase in the number of newly generated frequencies, which is not favorable in our case. In our work, we utilized a self-seeded approach to realize the nonlinear FWM process, where two optical waves of a wavelength within a visible (617 nm) and a visible blue light (470 nm) were generated as the Stokes and anti-Stokes beams, respectively. The wavelength conversion was realized within the IM-FWM process in an FMF nonlinear fiber section of a length equal 0.8 m. 

The experimental setup for the self-seeded IM-FWM process is presented in Figure 3.

The red-wavelength Stokes beam generated in one section of FUT was utilized as a signal beam and coupled with an intense pump beam (532 nm) to the second segment of the same type of fiber. Because both beams are pulsed, it is also crucial to match both pulses in the time domain. For that purpose, a pump beam coupled to the second segment of the FUT was routed through a delay line that compensates for the additional optical path taken by a signal beam (Figure 3).

## 3. Results

### 3.1. Single Pump Configuration

In the first part of the experiment, we characterized the optical field (spectral and spatial distribution) at the fiber’s output versus the power of a power pump. The generation of Stokes and anti-Stokes waves of wavelengths of 617 nm and 470 nm, respectively, is observed for the average pump power of about 5.0 mW. When the pump is not appropriately aligned, the FWM efficiency is significantly lower, and other spectral components or a mix of higher-order modal profiles can be visible for the Stokes/anti-Stokes waves. For the minimization of the optical power of a pump beam that results in the efficient generation of FWM spectral components, the position of the fiber was precisely re-adjusted with the use of piezoelectric actuators: first, to keep a clean Gaussian-like green spot of a pump Beam, visible in the far-field, as well as to maximize the power of the Stokes and anti-Stokes spectral components.

The spectra of the optical output field as a function of the pump’s power are presented in Figure 4a,b. The intense peaks centered at 617 nm and 470 nm can be clearly assigned to the FWM origin, as the Stokes and the anti-Stokes beams, respectively.

Figure 4c presents the power characteristics of the Stokes and anti-Stokes beams. The insets in Figure 4c show the intensity profiles of the low-power pump beam (left panel) and higher powers pump beam along with the Stokes and anti-Stokes waves (5.0 mW and 10.0 mW, middle and right panel, respectively) that are generated in nonlinear frequency conversion. They confirm that the pump beam remains in the fundamental LP_01_ mode while the FWM generated waves propagate in one of the HOM, which confirms an inter-modal character of the observed nonlinear process.

Almost 20% conversion efficiency was obtained in a single pump beam configuration. It was calculated as a percentage ratio of the Stokes and anti-Stokes waves powers at the output (measured using band-pass optical filters) to the pump beam power coupled to the optical fiber. However, based on the FWM efficiency versus input power characteristic presented in Figure 4d, it can be seen that further power increase above 8.0 mW of average power leads to saturation of the wavelength conversion efficiency. Instead of a higher Stokes and anti-Stokes beams power, a further increase in power may damage the end of the bare optical fiber. Additionally, increasing the length of the fiber section is not a good solution due to the significant increase in Raman scattering. As a method of increasing conversion efficiency, we verified experimentally the impact of additional seed beam on the process efficiency.

### 3.2. Self-Seeded Configuration

The performed self-seeded IM-FWM process consists of two parts. As presented in Figure 3, we used two pieces of an FMF fiber of the same type and the same length of about 0.8 m. Firstly, we obtained the Stokes and anti-Stokes beams of 617 nm and 468 nm, respectively, in the one part of FUT. Using the Pellin–Broca prism, a red beam was selected and utilized as a signal beam for the IM-FWM process realized in the second part of FUT. Such an approach guarantees that the wavelength and the modal profile of the utilized signal beam are automatically matched with the Stokes wave in the self-seeded process. Both beams, the signal (λ = 617 nm) and pump (λ = 532 nm), were secondly co-coupled to the second stage of FUT. The position of both fibers was carefully adjusted to maximize the power of the Stokes and anti-Stokes waves generated in both fiber sections to optimize the power of the signal beam and optimize the FWM overall conversion efficiency realized in the second segment of an FMF fiber.

As presented in the inset of Figure 5a, the modal profile of a signal beam indicates a mode LP_12_ that corresponds precisely to the mode number of a Stokes beam, generat-ed spontaneously in the second stage of FUT. The output field of a signal beam that propagates without an intense pump beam (inset of Figure 5a—left panel) indicates that the selective excitation of the higher-order mode (LP_12_) has not been achieved. Notwithstanding, the wavelength of a signal beam corresponds precisely to the Stokes beam and can be effectively gained along with the propagation in the second fiber. We think improvements in the signal beam coupling that would lead to the selective exci-tation of only the LP_12_ mode would enable even greater efficiency of the self-seeded IM-FWM process.

The presented approach is universal and provides good compatibility between the Stokes and the signal beams in the self-seeded IM-FWM process. It can also be applied to other types of optical fibers without introducing additional modifications to the experimental system. Due to the same modal profiles of the pump and signal beams, the efficiency of a seeded IM-FWM process would be significantly increased. 

Experimental results of the self-seeded IM-FWM are presented in Figure 5. The graph presented in Figure 5a compares the Stokes and anti-Stokes beams’ powers at the output of the second fiber power as a function of the power of the pump beam. In the case of the pump coupling only, the power of the Stokes (light red circles—Figure 5a) and anti-Stokes (light blue circles—Figure 5a) does not exceed the value of 1 mW. When a signal beam of the optical power of 40 μW only is coupled with the pumping beam, the efficiency of the FWM process increases significantly. The Stokes and anti-Stokes beams reach values above 1.5 mW and 2.5 mW, respectively. The comparison of FWM conversion efficiency is presented in Figure 5b. We can notice that the FWM threshold power decreases twice as a 40 μW signal beam propagates with the pump beam. The overall efficiency increases about three times, from 15% to almost 45%; however, a self-seeded FWM configuration (presented in Figure 3) can be optimized. The efficiency presented in Figure 5b was calculated in the same way as the results presented in Figure 4d, which means that the optical power needed to generate the signal beam is not included in the total power of a pump beam. When we calculate the total power of a Stokes and anti-Stokes beam with respect to the power of both pump beams, the overall efficiency of the self-seeded IM-FWM process is still significantly higher and is about 25%. 

## 4. Perspectives

FMF fibers have more freedom to design the fiber parameters than single-mode ones. The presented self-seeded IM-FWM configuration and the appropriate nonlinear optical fiber can be potentially applied in the construction of nonlinear optical frequency converters. Proper system design and utilization of the all-fiber technology could also contribute to more compact dimensions of the final device and increase the system’s stability. By managing the refractive index profile, we can directly affect the values of propagation constants for a given guided mode and meet the phase-matching condition for the desired wavelengths.

## Figures and Tables

**Figure 1 materials-15-04550-f001:**
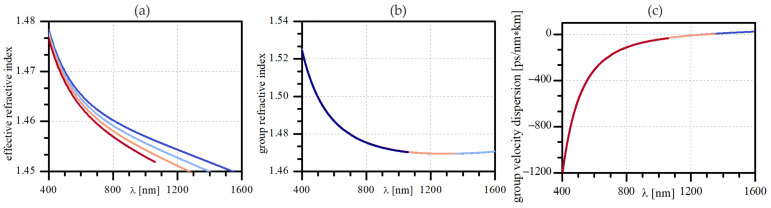
Spectral characteristics of analyzed few-mode fiber: (**a**) effective refractive index for fundamental mode (dark blue line) and a few higher-order modes: LP_11_ (blue line), LP_02_ and LP_21_ (orange line), LP_31_ and LP_12_ (red line); (**b**) group refractive index for LP_01_ and LP_11_ (blue line), LP_02_ and LP_21_ (orange line) and LP_31_ and LP_12_ (dark blue line); (**c**) group velocity dispersion for LP_01_ and LP_11_ (dark blue), LP_02_ and LP_21_ (orange line) and LP_31_ and LP_12_ (red line).

**Figure 2 materials-15-04550-f002:**
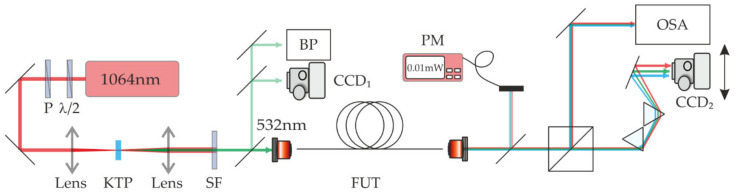
The experimental setup for the inter-modal FWM investigation in an FMF: P—polarizer, λ/2—half-wave plate, KTP—nonlinear crystal, SF—short pass filter, FUT—fiber under test, BP—beam profiler, PM—power meter, CCD—digital camera, OSA—optical spectrum analyzer.

**Figure 3 materials-15-04550-f003:**
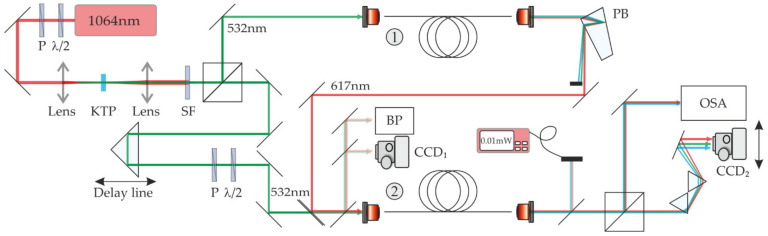
The experimental setup for investigation of the self-seeded IM-FWM process: P—polarizer, λ/2—half-wave plate, KTP—nonlinear crystal, SF—short pass filter, FUT—fiber under test, BP—beam profiler, PM—power meter, CCD—digital camera, OSA—optical spectrum analyzer, PB—Pellin–Broca prism.

**Figure 4 materials-15-04550-f004:**
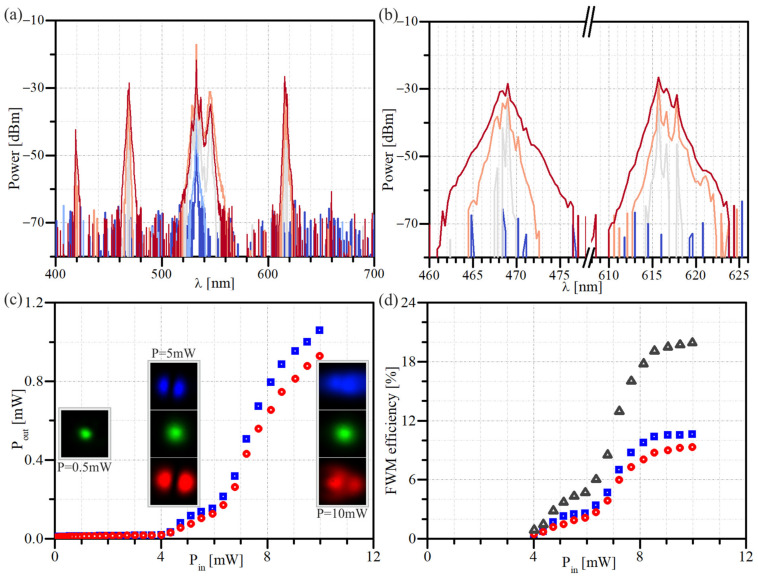
The experimental results of the FWM process realized in 0.8 m FUT: (**a**,**b**) output spectra for pump power 0.5 mW (dark blue), 2.0 mW (light blue), 5.0 mW (light grey), 7.0 mW (orange) and 10.0 mW (dark red); (**c**) power of the Stokes (617 nm—red circles) and anti-Stokes (470 nm—blue squares) beams; (**d**) wavelength conversion efficiency, plotted as a function of the power of the pump beam (532 nm); calculated for the Stokes (red circles), anti-Stokes (blue squares) and both Stokes (dark grey triangles) beams. The insets in (**c**) present the modal field distribution at the fiber output for 0.5 mW, 5.0 mW and 10.0 mW pump beam powers.

**Figure 5 materials-15-04550-f005:**
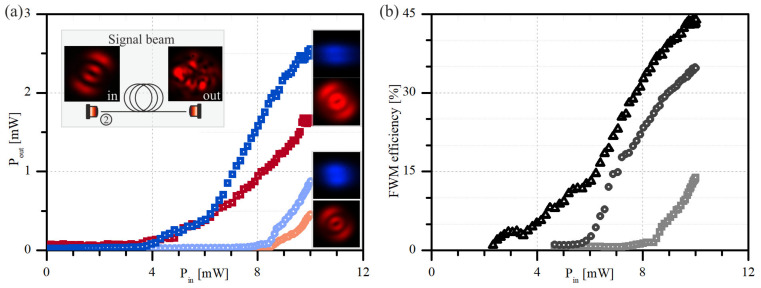
The experimental results of the FWM process realized in 0.8 m FMF fiber in the presence of a co-propagated signal beam: (**a**) output power of the Stokes (light red circles) and anti-Stokes (light blue circles) beams plotted as a function of the power of the pump beam for propagation without signal beam and together with a signal beam of a power of 40 μW (dark red and dark blue squares, respectively). The insets in (**a**) present the modal field distribution of a signal beam at the input and output of the fiber for signal beam propagation only (left panel), and Stokes/anti-Stokes modal field profiles obtained in the FWM process realized with and without signal beam of a 40 μW optical power (bottom and top right panels, respectively); (**b**) FWM wavelength conversion efficiency plotted as a function of the power of the pump beam (532 nm), for the signal beam powers equal 0 μW, 12 μW and 40 μW (light grey squares, grey circles and black triangles, respectively).

## Data Availability

Not applicable.

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
