# Peer review of "An Efficient Method for the Intermodal Four-Wave Mixing Process"

_materials, 2022, doi:10.3390/ma15134550_

Round 1
Reviewer 1 Report
The manuscript presents an efficient method for intermodal four-wave mixing process, which could significantly increase the wavelength conversion efficiency in FWM. There are some questions should be furtherly clarified by the authors.
- It is commonly known that for a particular FMF, the zero-dispersion frequencies of different supported spatial modes should be different. I see no relevant description of these critical parameters, which actually should be prevented.
- For the self-seeded configuration, from Fig. 4, we can see that signal beam looks like LP12, but no description mentioned how to generate the appropriate HOM, please add corresponding information. Moreover, the authors didn’t mention which HOM should be used and why. Please explain the reason.
- The current method of evaluating mode field purity is not good enough. S2 method should be good enough to give you both the accurate composition of mode field as well as the actual mode field distribution of each stimulated mode, which can also help you to further increase the stimulation purity.
Once the above questions are carefully addressed, I believe that the manuscript could be published in Materials.
Author Response
Dear Reviewer,
Thank you very much for all your valuable comments. In the revised version of the article, we addressed all of the above-mentioned issues. They concern in particular the following comments:
- "It is commonly known that for a particular FMF, the zero-dispersion frequencies of different supported spatial modes should be different. I see no relevant description of these critical parameters, which actually should be prevented."
ZDW for the fundamental mode and a few higher-order modes was provided. Also, a plot that presents exact characteristics of GVD relation versus wavelength was added to a fiber description section.
- For the self-seeded configuration, from Fig. 4, we can see that signal beam looks like LP12, but no description mentioned how to generate the appropriate HOM, please add corresponding information. Moreover, the authors didn't mention which HOM should be used and why. Please explain the reason.
Additional information was added to the text. The matching between the mode number of a signal beam and the Stokes beam generated spontaneously in the second stage of FUT is obtained automatically due to utilizing the same type and the same section length of FUT. We also put additional information on which set of pump/Stokes/anti-Stokes wavelength and modes we can obtain a phase-matching for the intermodal four-wave mixing process.
- "The current Method of evaluating mode field purity is not good enough. S2 method should be good enough to give you both the accurate composition of mode field as well as the actual mode field distribution of each stimulated mode, which can also help you to further increase the stimulation purity."
Information about the S2 method of evaluation mode field purity was added in the introduction. Of course, it is essential to determine the appropriate HOM for a signal beam. In the described experiment, the matching between the order number of HOM of a signal and Stokes beam was fulfilled naturally, as both beams were generated in the same nonlinear optical fibre. Of course, the phase-matching condition would be fulfilled in different types of optical fibre for different wavelengths/HOM; however, the desribed method still could be utilized without significant modifications to the experimental system.
I hope that the provided explanations will be sufficiently comprehensive and enable our work's publication.
Best regards,
Michal Kwasny
Reviewer 2 Report
In current submission, the authors demonstrate experimental results about the spontaneous IM-FWM and the self-seeded IM-FWM. It is obvious that, the efficiency of self-seeded IM-FWM is higher. The English presentation should be improved. Although the results showed that over 40% of the wavelength conversion efficiency in the FWM process can be obtained using a short fiber, the current version of the manuscript is not suitable for acceptance. Here are my comments,
1. Some expressions need to be rephrased, a few just as examples.
a) From line 30 to line 34, “In conventional step-index fibres, the phase-matching condition, among others, can be realized by pumping optical fiber by an intense beam close to the zero-dispersion wavelength [15], [16], birefringence matching [17], [18], or by realizing the so-called intermodal FWM (IM-FWM) process that requires a few-mode optical fiber (FMF) and allows for significant shifts between Stokes/anti-Stokes and fundamental beam(s) [8], [19].” It is a confused description. “among others”, what does “others” mean?
b) In line 75, “As a fiber section increases, the FWM can appear firstly”, it is better to modified as “As the fiber length increases,”
c) Other mistaken or redundant expressions are also observed in the manuscript, please check carefully.
2. In line 115, “two optical waves of a wavelength within a visible (617nm) and UV-band (470nm) were generated as the Stokes and anti-Stokes beams”, why that the light wavelength at 470 nm is a UV band, instead of a visible blue light?
3. All there are experimental phenomena without the theoretical analysis. It is better to provide some physical explanations. Both the inverse group velocity and the group velocity dispersion of the used FMF are helpful to evaluate the phase-matching condition, which play a vital role on the FWM efficiency.
4. The organization of this manuscript should be improved. It would be better to focus on the authors own job in Section 2 Materials and Methods. The description of the experimental setup in Section 3 Results would be better removed to Section 2.
Author Response
Dear Reviewer,
Thank you very much for all your valuable comments. In the revised version of the article, we addressed all of the issues mentioned above. They are concerned, in particular, with the following statements:
- Some expressions need to be rephrased, a few just as examples.
a) From line 30 to line 34, "In conventional step-index fibres, the phase-matching condition, among others, can be realized by pumping optical fiber by an intense beam close to the zero-dispersion wavelength [15], [16], birefringence matching [17], [18], or by realizing the so-called intermodal FWM (IM-FWM) process that requires a few-mode optical fiber (FMF) and allows for significant shifts between Stokes/anti-Stokes and fundamental beam(s) [8], [19]." It is a confused description. "among others", what does "others" mean?
b) In line 75, "As a fiber section increases, the FWM can appear firstly", it is better to modified as "As the fiber length increases,"
c) Other mistaken or redundant expressions are also observed in the manuscript, please check carefully.
The indicated sentences were rephrased. The context of the word "others" was not proper in this fragment. We also checked the manuscript carefully for the redundant expressions and made necessary corrections.
- In line 115, "two optical waves of a wavelength within a visible (617nm) and UV-band (470nm) were generated as the Stokes and anti-Stokes beams", why that the light wavelength at 470 nm is a UV band, instead of a visible blue light?
We changed a "UV band" for "visible blue light", as suggested.
- All there are experimental phenomena without the theoretical analysis. It is better to provide some physical explanations. Both the inverse group velocity and the group velocity dispersion of the used FMF are helpful to evaluate the phase-matching condition, which play a vital role on the FWM efficiency.
Additional charts presenting mentioned characteristics were added to the description of a fiber.
- The organization of this manuscript should be improved. It would be better to focus on the authors own job in Section 2 Materials and Methods. The description of the experimental setup in Section 3 Results would be better removed to Section 2.
The description of the experimental setup was moved to Section no. 2.
I hope that the provided explanations will be sufficiently comprehensive and enable our work's publication.
Best regards,
Michal Kwasny
Reviewer 3 Report
Review of the paper : Manuscript ID: materials-1727835
« An efficient method for intermodal four-wave mixing process”
This paper deals with experimental observations showing that the IM-FWM (Intermodal Four-Wave-Mixing) Stokes and anti-Stokes beams can be generated spontaneously from vacuum noise as well as during a stimulated process, where together with an intense pump beam, a weak signal beam of a wavelength matched to the Stokes beam is co-coupled in a few-mode nonlinear optical fiber in this case a weak signal beam is amplified through the fiber in the simultaneous presence of an intense pump beam, which results in a better wavelength conversion efficiency.. The authors analyze the Stokes and anti-Stokes beams power and the FWM process efficiency versus the power of a pump beam.
The authors propose by choosing a fiber length of 80 cm a kind of trade-off between efficient FWM frequency conversion to Stokes and anti-Stokes beams and a minimal degeneration of the FWM waves due to the Raman effect.
They used a self-seeded approach to realize the nonlinear FWM process, where two optical waves in the visible red (617nm) and UV (470nm) bands were generated as the Stokes and anti-Stokes beams, respectively. Subsequently, the red-wavelength Stokes beam was used as an additional signal beam and coupled along with an intense pump beam of wavelength 532nm (frequency doubling of Nd:YAG laser at 1064 nm) to another segment of the same-type of fiber. Because both beams, pump and signal, outputs temporal pulses, both pulses have been be matched in the time domain through a delay line that compensates additional optical path taken by the signal beam.
The experimental results show spectra of the output field as a function of the power of the pump beam with intense peaks centered at 617 nm and 470 nm corresponding to the FWM generated Stokes and the anti-Stokes beams. Power characteristics of the Stokes and anti-Stokes beams are also presented by the intensity beam spatial profiles and by the power of the FWM beams as a function of the power of the pump beam.
Experimental results also compare the Stokes and anti-Stokes beams powers at the output of the second fiber power as a function of the power of the pump beam. In the case of pump coupling only, the power of the Stokes and anti-Stokes do not exceed the value of 1mW. When a weak signal beam of 40 μW optical power is coupled with the pumping beam, the efficiency of the FWM process increases significantly and the FWM threshold power decreases. The overall efficiency increases about three times, from 15% to almost 45%.
The authors claim that the presented seeded IM-FWM configuration, together with the appropriate nonlinear optical fiber can profit the construction of nonlinear optical frequency converters. Proper system design using all-fiber technology could also contribute to more compact devices and increase system stability.
Author Response
Dear Reviewer,
Thank you very much for all your valuable comments. In the revised version of the article, we addressed all of the issues mentioned by all reviewers. Thank you for accepting the article and for your positive comments regarding the submitted manuscript.
Best regards,
Michal Kwasny
Round 2
Reviewer 2 Report
No more comments.